# Enhancing Privacy and Data Security across Healthcare Applications Using Blockchain and Distributed Ledger Concepts

**DOI:** 10.3390/healthcare8030243

**Published:** 2020-07-29

**Authors:** Haibing Liu, Rubén González Crespo, Oscar Sanjuán Martínez

**Affiliations:** 1Evergrande School of Management, Wuhan University of Science and Technology, Wuhan 430000, China; 2School of Economics and Management, Lanzhou Jiaotong University, Lanzhou 730070, China; 3Computer Science Department, School of Engineering and Technology, Universidad Internacional de la Rioja (UNIR), 26006 Logroño, Spain; ruben.gonzalez@unir.net

**Keywords:** data security, privacy, healthcare applications, blockchain technology, distributed ledger technology

## Abstract

Nowadays, blockchain is developing as a secure and trustworthy platform for secure information sharing in areas of application like banking, supply chain management, food industry, energy, the Internet, and medical services. Besides, the blockchain can be described in a decentralized manner as an immutable ledger for recording data entries. Furthermore, this new technology has been developed to interrupt a variety of data-driven fields, including the health sector. However, blockchain refers to the distributed ledger technology, which constitutes an innovation in the information recording and sharing without a trusted third party. In this paper, blockchain and Distributed Ledger-based Improved Biomedical Security system (BDL-IBS) has been proposed to enhance the privacy and data security across healthcare applications. Further, our goal is to make it possible for patients to use the data to support their care and to provide strong consent systems for sharing data among different organizations and applications, since this includes managing and accessing a high amount of medical information, and this technology can maintain data to ensure reliability. Finally, results show that new blockchain-based digital platforms allow for fast, easy, and seamless interactions between data suppliers to enhance privacy and data security, including for patients themselves.

## 1. Introduction

Recent trends in technology are exploited for diverse real-world applications to provide definite solutions for end users. Assimilating technological aspects in user-related application provides diverse advantages, from the quality of service (QoS) to security [1]. The healthcare platform is visualized using electronic health records (EHRs) in its digital and technical format, providing unrestricted access to the end users. Diagnosis centers and healthcare infrastructures provide different access and data sharing processes for their users through EHRs [2,3,4]. EHR is an organized set of patient-/user-related information that is digitally shared through a secure platform for ubiquitous access [5]. User applications and graphical user interfaces designed for EHR access provide access to the healthcare data through simple authorization and authentication procedures. Since sensitive information, end-to-end security, and privacy are the prime concerns in sharing EHR’s between users [6], this is vital as the technology requires additional infrastructures such as cloud, Internet of things, mobile devices, etc. for sharing EHR’s [7]. 

Blockchain is another technology that is commonly used in different applications for providing distributed access to resources and unalterable information [8]. The blockchain paradigm is used for administering security in different communicating and processing systems. Healthcare application does not require trusted third-parties for administering security [9]. The electronic ledger is distributed across different communicating and processing systems to improve the swiftness in security administration and privacy preservation [10]. Besides, blockchain eases EHR sharing between end-user applications and healthcare infrastructures without interrupting the communication process [11,12]. Such facilities are provided through line-of-trust and authentication with interoperability using the distributed electronic ledger technology. Modern healthcare applications concentrate on the privacy of the users and security of the information shared to prevent anonymous and unauthorized access to illegitimate users [13,14]. 

Trust, authentication, and privacy are the major requirements in sharing EHRs between different users. Administering the blockchain paradigm as a decentralized ledger for monitoring shared information is becoming a familiar practice in recent years [15,16]. Blockchain-assisted authentication and trust-based security are assimilated with the medical systems for improving the quality of information sharing and preventing unauthorized interruptions [17,18]. Knowing the significance of the data, biomedical systems rely on robust authentication and trust schemes for confronting diverse attacks, data leakage, tampering, and loss. EHR access control, defining security levels, verifying users, and sharing sessions are collaboratively performed using the security systems [15,17,19]. Modified and sophisticated access control, encryption/decryption schemes, and auditing features are required to handle different attacks and illegitimacy in storing and sharing EHRs. In trust-based schemes, user-centric factors are assessed to differentiate the users to provide access controls, whereas authentication schemes focus on providing data/EHR security through hashing and encryption/decryption process [20,21]. 

However, blockchain refers to the distributed ledger technology, which constitutes an innovation in the information recording and sharing without a trusted third party. In this paper, Blockchain and Distributed Ledger based Improved Biomedical Security system (BDL-IBS) has been proposed to enhance the privacy and data security across healthcare applications.

## 2. Related Works

Tang et al. [22] proposed privacy-preserving healthcare in the trusted network to enhance the trustiness among the patient and caregivers. The Sybil attack is used to find the fake patient and terminate it from the network. The proposed method is used to make the authenticated person access the healthcare center.

Computer-aid design is implemented for security, and privacy of the trusted systems is introduced by Salnitri et al. [23]. It also gives the specification of experts to use the system from various characteristics. They are also using the higher goal for the business, and external threats are maintained for the trustworthiness in the network.

S-Alex convolution neural network and dynamic game theory (SCNN-DGT) designed by Kong et al. [24] are used in the IoT-cloud computing environment for health data management. The initial step is obtaining the information of the healthcare and classifying them in Alex’s net convolutional network. This method is designed to evaluate security in the healthcare system. It validates the index screening to verify the user.

Data integrity is used for sharing the records of healthcare in a verifiable way and is introduced by Wang et al. [25]. The author developed a blockchain for privacy usage through symmetric encryption and attribute-based encryption. It attains the fine-grained access control.

Zhao et al. [26], developed key management for healthcare blockchain. The efficient key management method is used as a privacy and security mechanism in the healthcare system. It is observed by embedding the sensor to analyze the blockchain. The proposed method is used to enhance the effectiveness and high security. 

Guo et al. [27] modeled a multi-authority for the Tele-medical system to improve the efficient blockchain based on the ABE scheme. In this paper, both the dynamic authentication and authorization are used for MoD service under telemedicine. ABE is mainly used to manage the system in real-time scenarios for private healthcare data. This is done in a cloud-based environment.

A blockchain is proposed for the medical records to access and permits the MedChain process, which is addressed by Daraghmi et al. [28]. Medchain is used for interoperating, secure, and effective access for patients’ privacy. The security is time-based access that gives the degree of health providers. 

A blockchain is used for the Electronic Health Record system (EHRs) and is proposed by Guo et al. [29]. The authors implemented a secure attribute based on signature with multi authorities. The patients send the text according to the health as the attribute evidence to the healthcare center. The trust is given to the authorities to access the message, and both use the public and private keys to avoid the escrow problem.

The medical service framework is designed to store the secure records of the patient by using the blockchain method and is introduced by Chen et al. [30]. The storage is done on the cloud for large data access. The records are shared by its aspect based on its service related to the authorized user.

Tian et al. [31], observed medical data management with private access. The blockchain is used to protect the data in two aspects such as storing the data in the local database, encrypting the data, and sharing the key to the patient for further viewing. The shared key for security and integrity is established using sibling intractable function families (SIFF) aided by blockchain. The proposed method uses integrity, availability, and privacy of medical data for better efficiency.

Wang et al. [32] presented an e-healthcare system by using Wireless Body Area Networks WBAN. The blockchain is used to generate security and resolve the low power healthcare system. The WBAN is placed in the patient’s body and transmits the data by using the blockchain process.

A blockchain -based healthcare system using formal methods is developed by Brunese et al. [33]. This paper aims to exchange information from the patients to the hospital network by using magnetic resonance images. The data are transmitted by the formal equivalent for validation. They are modeled by radiomic features for automata.

Uddin et al. [34] proposed blockchain leveraged decentralized eHealth architecture (BDeHA). This architecture consists of three layers, including a sensing layer for obtaining the data through the sensor. The second is NEAR processing for sensing the IoT devices and the third one is FAR processing, which is comprised of cloud computing servers.

Griggs et al. [35] observed a healthcare blockchain using smart contracts for patient monitoring. The smart contracts are used for secure analysis management for communication with the sensor. They are also used to monitor the patients and professionals to give notification regarding the health.

Brodersen et al. [36], globally and across several industries, present an innovation model that will allow business to business-and-consumer transactions to be faster, more efficient, and highly secure. Many healthcare participants hope the same distributive database technologies allowing this new model can lead to similar outcomes within the industry and recognize that confusion, like many other major innovations.

## 3. Blockchain and Distributed Ledger Based Improved Bio-Medical Security System

The proposed BDL-IBS is designed to improve the trust- and privacy-related specifications of the electronic shareable health records. The system focused on maximizing the sharing rate of the secured records along with less adversary impact. In this system, blockchain technology is exploited by the medical server that tracks the trust privacy factors between the users and records. In Figure 1, an illustration of a biomedical security system with blockchain technology is presented.

The components of the bio-medical system include storage and a medical server. The storage contains the health records of the end-users in a digital format. The medical server is responsible for processing user requests and responding to them with appropriate records. A common sharing platform such as cloud and associated infrastructures are responsible for sharing EHRs. The blockchain and distributed ledger are used in both the medical server and end-user applications. In the blockchain associated with the medical server, the trust and privacy factors are analyzed, whereas the privacy factors are alone assessed in the end-user blockchain. The trust factors include successful access and response to request ration, and privacy relies on convergence and complexity. The trust process is analyzed and explained in detail in the following subsections.

Adversary Model: In this bio-medical security system, malicious access due to man-in-middle and data tempering adversary models are considered. In a man-in-middle attack, the adversary overlaps the end user to gain access to the HER. This results in sharing health information to an adversary and thus degrading the design of a secure biomedical system. In the case of a data tempering attack, the adversary breaches HER from any node communicating with the biomedical system. It either modifies the actual data/tracks the communication through the HER information. Figure 2a,b portrays the representation of the man-in-middle and data tampering attacks over the EHR.

For thwarting the above attack, the trust model and concentric authentication are introduced using the blockchain paradigm. As referred to earlier, the blockchain process is differentiated in both the medical server and end-user functions. 

Apart from the regular two-layer network, the man-in-middle attack can be overcome by the server-client based blockchain technology as shown in Figure 2c. Since it is a server–client network, it is well suited for the medical user and end-user functions. To reduce the man-in-middle issue, a pure application-oriented implementation is followed in the objective of the proposed idea. A proper set of protocols should be determined in the server domain, and the appropriate application receives the data from the client side.

The process of trust-based validation is performed using linear decision-making, and authentication is augmented through classification-based learning.

Trust model based on Linear Decision Making: In the trust model, the factors are successful access and end-user application to fetch HER. Through conventional communication standards, the end-user application generates a query for accessing HER. The initial authorization for the end-user is provided using login ID/name and password information. This information is validated by the medical server to ensure the reputation of the user. The medical server is associated with the blockchain with the following entries, as in Table 1.

For each Q generated and received in the medical server, the state of R (i.e., sharing EHR), the factors $c,\;t_{s},\;t_{v},$ and τ are updated. This information remains unchanged in the blockchain paradigm. It is to be noted that τ is valid for $t_{v}$, within which the sharing of EHR is completed. For any case of $t_{v}<t_{s}$, the τ→0 and the user is marked as illegitimate. For validating the above conditions, τ is computed as a linear combination of (R, Q) and successful access probability $\left(\rho_{a}\right)$. In a given $t_{v}$, the τ is computed as
(1)$$\begin{array}{c} \tau \left(t_{v}\right)=\frac{R\left(t_{s}\right)}{Q}+\rho_{a}\\ where,\;\\ \rho_{a}=\left(\frac{c}{Q}\right)+\left(1-\frac{R}{Q}\right)R \end{array}\}$$

The factor $\frac{R}{Q}$ is the ratio of response to the query request received by the medical server. The linearity in identifying the trust for a period of $t_{v}$ relies on $\frac{R}{Q}$ and $\rho_{a}$, where both the factors are proportional to each other. The above linear relationship between $\rho_{a}$ and $\frac{R}{Q}$ is $t_{s}$ is recurrently analyzed using the $\frac{t_{s}}{c}$ instance, i.e., the τ in all c instances is verified from its previous shared count that is given as
(2)$$\begin{array}{c} \bar{\tau }=\frac{1}{\bar{c}}\left[\frac{R_{1}}{Q_{1}}\left(\frac{t_{s}}{1}\right)-\rho_{a}\left(\frac{t_{s}}{2}\right)+\frac{R_{2}}{Q_{2}}\left(\frac{t_{s}}{2}\right)-\rho_{a_{2}}\left(\frac{t_{s}}{2}\right)+\cdots +\frac{R_{c}}{Q_{c}}\left(\frac{t_{s}}{c}\right)-\rho_{a_{c}}\left(\frac{t_{s}}{c}\right)\right]\\ =\frac{1}{\bar{c}}\left[\overset{c}{\underset{i=1}{\sum }}\frac{R_{i}}{Q_{i}}\left(\frac{t_{s}}{i}\right)-\rho_{a}\left(\frac{t_{s}}{i}\right)\right]=\frac{t_{s}}{c}\left[\overset{c}{\underset{i=1}{\sum }}\frac{1}{i}\left(\frac{R_{i}}{Q_{i}}-\rho_{a}\right)\right] \end{array}\}$$

From the above sequence, the varying $\frac{R}{Q}$ or $\rho_{a}$ in $t_{s}$ is estimated for all the c shared to the end-user. In this sequence, the varying point p initiating the change in proportionality between $\rho_{a}$ and $\frac{R}{Q}$ is identified. Such identification helps to reduce the computations and security mechanisms (authentication) to prevent losses in sharing EHR. This point from the sequence $t_{s}$ is computed using Equation (3) as
(3)$$p=\overset{c}{\underset{i=1}{\sum }}\frac{\left[1-\tau {\left(t_{v}\right)}_{i}\right]}{\left[{\bar{\tau }}_{i}-\tau {\left(t_{v}\right)}_{i}\right]}$$

This validating point helps to hold the verification process and trust update in the blockchain, where the actual c is updated until $p\in \;\bar{\tau }$ sequence. The decision for pursuing/halting EHR sharing is determined using the conditions formulated in Table 2.

The last three conditions in Table 2 represent the unfeasible conditions as $\bar{\tau }<\tau \left(t_{v}\right)$ results in a negative p that is not possible in case c>1. Similarly, the sequence and instant trust are the same in case of sharing only 1 record, after which p=∞. This provides continuous chances for EHR sharing, whereas, in practical EHR based biomedical systems, the condition does not hold. For p≥c condition, the point is detected after all the counts are shared. Therefore, the previous state of name/ID for which it is τ with the new $t_{s}$ or $t_{v}$ period. The blockchain is updated for the above and hence for further sharing of EHRs. The case of the first two conditions is different, where p<c follows $\bar{\tau }$ and$\;\tau \left(t_{v}\right)$ as in Equations (3) and (2), respectively. The different case of condition 1 is to be differentiated from the other conditions as a trial to the user is given if the current trust is less than the previous sequence of trust. This impacts either $\rho_{a}$ or $\frac{R}{Q}$ and hence Equation (1) is modified as
(4)$$\tau \left(t_{v}\right)=\{\begin{array}{c} \left[1-\frac{R\left(t_{s}\right)}{Q}\right]+\rho_{a}\left(\frac{t_{s}}{c}\times t_{v}\right),\;if\;\frac{R}{Q}\;isnotaconstant\\ \frac{R\left(t_{s}\right)}{Q}+\frac{Q-R}{Q}\rho_{a},\;if\;\rho_{a}\;isnotaconstant \end{array}$$

If both the $\frac{R}{Q}$ and $\rho_{a}$ factors are not constant, then the sharing process is halted. Based on the different instances for $\frac{R}{Q}$ (or) $\rho_{a}$, the decision is made such that the sharing is not halted, whereas it is paused until the next update if τ is observed. In this pausing instance, the sharing session of the end-user application is expired. Therefore, the user has to login again to re-initiate the EHR sharing session. The time of validity based on different instances of$\;\tau \left(t_{v}\right)$ is determined using Equations (5) and (6), respectively.
(5)$$\begin{array}{c} t_{s_{1}}=t_{v}-\left(\frac{Q_{1}-R_{1}}{Q_{1}}\right)t_{s}=t_{v}\left(ast_{s}=0\;for\;the\;first\;instance\right),\;\forall \;\left(1-\frac{R}{Q}\right)<\rho_{a}\\ t_{s_{2}}=t_{v}-\left(\frac{Q_{2}-R_{2}}{Q_{2}}\right)\frac{t_{s}{}_{1}}{2}\\ \begin{array}{c} \vdots \\ t_{s_{c}}=t_{v}-\left(\frac{Q_{c}-R_{c}}{Q_{c}}\right)\frac{t_{s}{}_{c}}{c} \end{array} \end{array}\}$$
(6)$$\begin{array}{c} t_{s_{1}}=t_{v}-\left(1-\frac{R_{1}}{Q_{1}}\right)\rho_{a}t_{s_{o}}=t_{v}\;,\;\forall \;\left(\frac{Q-R}{Q}\right)\rho_{a}<\frac{R}{Q}\\ t_{s_{2}}=t_{v}-\left(1-\frac{R_{2}}{Q_{2}}\right)\rho_{a}t_{s_{1}}\\ \begin{array}{c} \vdots \\ t_{s_{c}}=t_{v}-\left(1-\frac{R_{c}}{Q_{c}}\right)\rho_{a}t_{s_{c}} \end{array} \end{array}\}$$

For the above Equation of computing $t_{v}$ for fluctuating $\tau \left(t_{v}\right)$, tn Figure 3a,b, respectively.

The process of trust-based update in the blockchain is performed using $t_{s}$ using $\frac{R}{Q}$ and $\rho_{a}$ factors independently. The process is consecutive if $t_{s}$ and $t_{v}$ is updated based on $\frac{R}{Q}$ and concurrent if the update is based on $\rho_{a}$. The process of differentiation relies on the p that is identified for both the conditions where ${\left(\frac{R}{Q}-\rho_{a}\right)}_{i}<{\left(\frac{R}{Q}-\rho_{a}\right)}_{i-1}\;\forall \;i\in c$. Finally, the user with max{τ} or $\max\{\left\{\bar{\tau }\right\}\left(t_{s}\right)\}$ is identified in all the instances for providing better authentication. The linear representation in Equation (2) is either fluctuate between $t_{s}$ based on $\frac{R}{A}$ and $\rho_{a}$ independently. The fluctuation is based on the varying $t_{s}$ and $t_{v}$ instances as differentiated by p. This trust-based decision-making helps to improve the ratio of successful sharing under controlled response time. In Table 3, the observed records that are classified under different conditions of Table 1 is presented for the different sharing times.

There is only one ending transmission in the sharing time of 70, where condition 4 is satisfied by sharing count of c. The records classified under conditions 3/5 are not sent to the end user, and hence their sessions are logged out. 

## 4. Classification-Based Concentric Authentication 

In the classification-based concentric authentication, EHR is shared. In a concentric authentication, the common classification on point p serves as the decision-making for generating authentic records. The classification-based learning allocates two types of non-sequential session keys for authenticating the sharing session. This classification is based on the fluctuating τ(tv) as in Equation (4). The impact of either of the fluctuation varies the administration of session keys to prevent the data tampering attacks. Initially, the session is set up between the medical server, and the end-user application follows a linear mapping map: RX Rc→ℝU. Here, Rc is the group of response until a count c, and ℝU is the random function of the end-user (U). The group consists of a random generator r∈Rc along with a differential prime number pn. For the different τ(tv), the variable r∈Rc relies on computing hashes HMS and HU for the medical server and end-user, respectively. The general format of an initial authentication is denoted as {Rc,RU,pn,r, HMS,HU,c}. The shared record count is obtained from the blockchain, where the trust of user access coupled with the records is stored. The distributed access to blockchain stored information is assessed in both end-user and medical server levels. For this authentication process, the classification occurrences of (1−RQ) and ρa in ts is performed. As stated previously, the sequential and concurrent update of the medical server blockchain process requires different session keys and authentication procedures. Therefore, the occurrence of p for condition 1 from Table 2 is the determining factor. Let ρp and ρs represent the fluctuating and sequential probabilities in a given time ts; then,
(7)such that ρ(s|p)=ρ(p|s)ρsρpρ(p|s)=∏i=1cρ(pi|s)}

$\begin{array}{c} \mathrm{As}\;\rho \left(s|p\right)=\frac{\prod_{i=1}^{c}\rho \left(p_{i}|s\right).\rho_{s}}{\rho_{p}}=\frac{\prod_{i=1}^{c}\rho \left(p_{i}|s\right)\rho_{s}}{\left(1-\rho_{s}\right)} \end{array}$, the above classification of probability, s over p is computed for all $\rho_{s}$ instead of $\rho_{p}$ to linearize the solutions as in Equation (1). Based on the relationship between $\frac{R}{Q}$ and $\rho_{a}$, the classification of ρ(p|s) is performed as
(8)$$\left(or\right)\begin{array}{c} \rho \left(s|p_{1},\;p_{2},\ldots p_{c}\right)=\frac{\rho_{s}\;\rho (p_{1},p_{2},\ldots p_{c}|s)}{\rho \left(p_{1},p_{2},\ldots p_{c}\right)}\\ \rho \left(s_{1},s_{2},\ldots s_{c}|p\right)=\frac{\rho \left(p|s_{1},s_{2},\ldots s_{c}\right)\rho_{p}}{\rho_{p}}=\rho (p|s_{1},s_{2},\ldots ,s_{c}) \end{array}\}$$
where $\rho \left(p|s_{1},s_{2},\ldots s_{c}\right)=\frac{\rho_{p}\prod_{i=1}^{c}\rho (p|s_{i})}{\rho \left(p_{1},p_{2},\ldots p_{c}\right)}.$ For condition 1, the classification rule is framed as in Equation (9) for identifying p over s as in Equation (8)
(9)$$\begin{array}{c} \rho (p|s_{1},s_{2},\ldots s_{c})≃\rho_{p}\overset{c}{\underset{i=1}{\prod }}\rho (p_{i}|s) \end{array}\}$$
where $\begin{array}{c} s=\underset{c}{\mathrm{argmax}}\rho_{p}\prod_{i=1}^{c}\rho (p_{i}|s) \end{array}$. Here in Equation (9), the probability of $\rho_{p}$ is computed based on the likelihood of p’s instances and its normalization as
(10)$$N\left(p\right)=\frac{c\times \rho_{p}+s}{\rho_{p}+\left(c\times s\right)}$$

The above likelihood normalization of p helps to classify$\;p\in \left(1-\frac{R}{Q}\right)<\rho_{a}$ condition or$\;p\in \rho_{a}<\frac{R}{Q}$ condition. This helps to decide between sequential and concurrent authentication procedure through the same concentric point from the fluctuating sequence of $t_{s}$. The normalization identifies precise p in the series of ρ(p|s) such that ρ(s|p) follows sequential authentication, whereas the previous occurrence relies on random concurrent security measures. Here, the priority of authentication is initiated from the first occurrence of $\rho_{p}$ of $\rho_{s}$ as determined by N(p). For all the first occurrences of $\rho_{p}$ and $\rho_{s}$, the sequence follows $\rho (p|s_{1},s_{2},\ldots s_{c})$ or $\overset{c}{\underset{i=1}{\prod }}\rho (p|s_{i}),$ and $\frac{\prod_{i=1}^{c}\rho \left({p|s}_{i}\right)\rho_{s}}{\left(1-\rho_{s}\right)}$ (as in Equation (6)). Using this sequence and concurrency, the authentication is presented as follows. In two cases, the occurrence of the sequence and concurrency observed is discussed below.

Case 1: The sequence initiates with $\rho_{s}$

Analysis 1: The hash sequence for both $H_{MS}$ and $H_{U}$ is formulated as
(11)$$\begin{array}{c} H_{MS}\left(p\right)=r^{i}\left|pn\right|+r^{i-1}\left|pn\right|+\ldots +r^{i-c}\left|pn\right|p^{c-1},\;\forall \;i\in c\\ and\\ H_{U}\left(c\right)=r^{i-c}\left|pn\right|+r^{i-c+1}\left|pn\right|+\ldots +r^{i}\left|pn\right|\rho \left(p|s_{i}\right),\;\forall \;i\in p \end{array}\}$$

This hash is composed of $\left[R_{c},{\;H}_{U}\left(p\right),\;c\right]$ and $\left[R_{U},{\;H}_{u}\left(c\right),\;c\right]\;\forall \;\left\{R_{c},{\;R}_{U},pn,\;r,\;c\right\}$ and is subject to verification using the user ID and session key as follows,
(12)$$\begin{array}{c} K_{s_{ij}}=H_{MS}\left[H_{U_{j}}\left(Id\right)\right]\left|pn\right|+r^{i-j\;},\;\forall \;i\in c\;and\;j\in p\\ and\\ K_{v}=\overset{c}{\underset{i=1}{\prod }}g^{i}\left|pn\right|-\left(i-p\right) \end{array}\}$$
where $K_{s}$ and $K_{v}$ are the secret and verification keys generated for the hashes, and therefore in the sharing process, $K_{s}\left[H_{MS}\left(p\right),\;R,c\right]$ is contributed to the end-user. At the receiver end, the $K_{v}$ is used for verification. If the process of sharing the records is sequential, then i∈c is sequential until p or the likelihood N(p) occurs. This is followed for all $\left[H_{MS}\left(p\right),R\right]$ until the c=p is reached, and then the coherency of $H_{U}\left(c\right)=H_{U}\left(p\right)$ until ρ(s|p) is observed. The verification of the process is also sequential by mapping $R\times R_{1\;\mathrm{to}\;p}\to R_{U}$ where $R_{U}$ is observed from the range of hashes from 1 to $\rho (s|p_{1},p_{2},\ldots {\;p}_{c})$. The first sharing verification is performed as
(13)$$\begin{array}{c} {}[H_{MS}\left(1\left|\right|B),r\right]=[H_{MS}\left(1\left|\right|B),\;K_{s}\right]\\ {}[H_{MS}\left(2\left|\right|B),r\right]=[H_{MS}\left(2\left|\right|B),\;K_{s}\right]\\ \begin{array}{c} \vdots \\ {}[H_{MS}\left(p\left|\right|B),r\right]=[H_{MS}\left(p\left|\right|B),\;K_{s}\right] \end{array} \end{array}\}$$
where, B denotes the blockchain record for the grouped storage of [R,c] after the hashing process. In the verification at the user end, the relevance is first validated, followed by the verification process as in Equations (14) and (15) respectively.
(14)$$H_{MS}\left(p\left|\right|B),\;r\right]=\{\begin{array}{c} \begin{array}{c} \left[H_{u}{\left(p||B\right)}^{pn},\;r\right]\\ \left(or\right) \end{array}\\ \begin{array}{c} {}[H_{u}(p||B),\;r^{c-p}\\ \left(or\right) \end{array}\\ \left[H_{u}\left(p\left|\right|B),\;c\rho \left(s_{i}|P\right)\right],\;i\;\in c\right) \end{array}$$
(15)$$\left[\frac{\prod_{i=1}^{c}\rho \left(p_{i}|s\right)\rho_{s}}{\left(1-\rho_{s}\right)},\;H_{MS}(P||B),\;K_{s}\right]=\left[\overset{c}{\underset{i=1}{\prod }}B_{i}.H_{U}{\left(Id\right)}_{i}K_{v_{i}},\;r\right]$$

In the above, the range of c is valid until p, i.e., the N(p) is the halting factor for sequential authentication. In the verification process, sequence as mapped in $R\times R_{C}\to R_{U}$ is the balancing factors where the sending and receiving sequence until ρ(p|s) is obtained. In this case, the converging interval of the proposed method is extended until the c, i.e., the restricted time from 1 to p is extended from p to c in a concentric manner. The next sequence for p to c authentication is discussed in Case 2.

Case 2: The sharing sequence experiences $\rho_{p}$.

Analysis 2: This case is unique as both sequential and concurrent authentication is performed with interfering with other processes. It is to be noted that the convergence time from the sequential process is experienced to $\rho (p|s_{1},{\;s}_{2},\ldots s_{c})$ from the $\rho_{p}$. This helps to identify more ρ(s|p), and thus the concentricity of the authentication process is expanded, reducing the chances of convergence. In this authentication process, both $H_{MS}$ and $H_{U}$ are used for performing secure sharing between the medical server and the end user. The blockchain is updated with p and N(p) along with the previous sequence for the appropriate user ID. Therefore, the session is initiated by verifying the following
(16)$$\begin{array}{c} {}[H_{MS}\left(p\left|\right|B),\;\left(c-p\right)\right]=[H_{MS}\left(p\left|\right|B),\;K_{s},\;p\right],\;\forall \;p\;to\;c\;in\;the\;medical\;server\\ and\\ {}[H_{U}\left(c-p\left|\right|B),\;c\right]=[H_{U}\left(c-p\right)||B,\;K_{v},c],\;\forall \;preceived\;by\;the\;end\;user\;application \end{array}\}$$

There are two verification steps followed for authenticating the sharing due to the fluctuating instances in $t_{s}$. The first authentication follows Equation (14), whereas the range from p to c follows
(17)$$\begin{array}{c} {}[H_{MS}\left(p\left|\right|B),\;c-p\right]=\{\begin{array}{c} \left[H_{MS}{\left(p||B\right)}^{c-p},\;c\right]\\ \left(or\right)\\ H_{MS}\left(c-p\right)||B,\;\rho \left(p|s_{i}\right),\;i\in c \end{array}\\ and\;\\ {}[H_{U}\left(c-p\left|\right|B),\;\rho_{p}\right]=\overset{c}{\underset{i=1}{\prod }}B_{c-i}H_{U}{\left(Id\right)}_{i}K_{v}.{\left(v-p\right)}_{i} \end{array}\}$$

The above process of authentication in sharing and receiving B is performed in both the medical server and the end user. Finally, the received B is verified using 1 to p sequence as in Equation (15), whereas the t to c received B is verified as follows.
(18)$$[H_{MS}\left(\rho \left|\right|B),\;\left(c-p\right),\;K_{s}\right]=\left[H_{u}\left(c-p\left|\right|B),\;H_{U}\left(Id\;\right),\;K_{v},\;c-p\right)\right]\forall \;i\;\in p\;to\;c$$

This verification is processed for all the fluctuating shared R through the classification process. This prevents unnecessary convergence and overload complexity in handling medical records at different time instances. In Table 4, the $\rho_{s}$ and $\rho_{p}$ for the varying p in different sharing time along with the complexity is tabulated.

In Table 4, the complexity is computed as the number of additional hashes generated due to $\rho_{p}$ to the actual existing hashes. The complexity is measured in terms of count of additional steps required for verification and authentication as observed in the keying process. If the impact of attacks is high, then the $\rho_{p}$ factor increases to prevent unnecessary data tampering or modification. Hence, in this case, the number of c fluctuates as the classification is grouped under both the sharing instances. 

## 5. Performance Analysis

The performance of the proposed BDL–IBS is assessed using simulations using an opportunistic network environment. In this environment, a maximum of 100,000 EHRs (unique and repeated) are shared for 110 users in different time instances. A user is capable of generating four Qs at the same time, for which the sharing interval is 90 s (max). The medical server of storage 4 × 1 TB is used for storing LHRs, and two blockchain servers with restricted read/write access are configured in this simulation environment. The medical server is capable of dispatching 20 records of size 70 mb in 1 s time. The maximum wait time for a record is 60 s, and the hash process follows hyperelliptic curve cryptography of a maximum size of 160 bits. Similarly, $K_{v}$ and $K_{s}$ is fit as 48-bits and 36-bit, respectively. Using this simulation environment, the existing 31FF [23], BDe HA [26], and SCNN–DGT [16] methods are considered for comparative analysis. For this comparative analysis, the metrics sharing ratio, response time, computation time, and convergence time are analyzed.

### 5.1. Successful Sharing Ratio

The proposed security system relies on record—user-access-based trust and differential authentication to improve the successful sharing of EHRs. The trust-based relationship between $\rho_{a}$ and R/Q is validated for the possible conditions in Table 2, generating $\tau \left(t_{v}\right)$ and $\bar{\tau }$ at different instances. In the sharing instances, pursuing/pausing sharing is determining based on $\rho_{a}>\frac{R}{Q}$ or $\rho_{a}<\frac{R}{Q}$ conditions. This condition-based decision-making determines $t_{s}$ for (p+1) to c instances and or $t_{s}$ for$\;\left(\frac{p+1}{2}\;\mathrm{to}\;c\right)$ instances in either sequential/concurrent manner. The concentric sharing process follows $t_{s_{c}}$ for any instance of $\tau \left(t_{v}\right)$; if the $\tau \left(t_{v}\right)$ is maximum, then the sharing is performed either in a sequential or concurrent manner. In this process, the blockchain updates the trust for the linear $\rho_{a}$ and $\frac{R}{Q}$ relation, which remains unchanged. Therefore, sharing for varying time and EHRs follows conditional satisfaction as in Table 2, achieving a high successful sharing ratio (refer to Figure 4a, b).

### 5.2. Response Time

The sharing time $t_{s}<t_{v}$ is ensured in all the instances of EHR processing for the received *Q*. If $t_{v}<t_{s}$ is observed, then the response time increases. For analyzing the instances of sharing c, the variable $\bar{\tau }$ and $\tau \left(t_{v}\right)$ is differentiated. In this case, $t_{s_{c}}$ for $\rho_{a}>\frac{R}{Q}$ is estimated as $t_{v}-\left(\frac{Q_{c}-R_{c}}{Q_{c}}\right)\frac{t_{s}{}_{c}}{c}$ and $t_{v}-\left(1-\frac{R_{c}}{Q_{c}}\right)\rho_{a}t_{s_{c}}$ independently. If the condition $t_{s}<t_{v}$ is achieved, then the varying point p is identified to differentiate the sharing of EHRs. Therefore, the joint sharing is not facilitated for trust varying or condition 1 (Table 2), dissatisfying users. Hence, a small wait time in a response is experienced; this disintegrates the conditions of $t_{s}<t_{v}$, where concurrent sharing and authentication is performed without additional wait time. Therefore, for the conditions 1 and 2, the response time for a *Q* from the end user is less compared to the other methods (refer to Figure 5).

### 5.3. Computation Time

Figure 6 presents the computation time of the proposed system as a comparative analysis with the existing methods. The authentication computing process requires either of the instances based on p, from which HMB and Hu are commonly adapted for the varying impact of untrusted users (classified under conditions 3 and 5 from Table 2). This helps to process the same number of c with the different authentication process and thereby reduces the complexity and required computations in the sequential sharing. Instead, the concurrent dissemination process of the records requires a change in first-level authentication as Equations (12) and (17) to satisfy N(p), confining $t_{s}$ within $t_{v}$. Therefore, the required computation increases by 1, and hence some additional time for verifying the second authentication is required. The verifying process is common in both the instances, demanding less/same time of computation. Hence, the overall computation time is differentiated by $\rho_{p}$, and ρ(p|s) and ρ(s|p) is less in the proposed security system.

### 5.4. Convergence Time

The proposed security system achieves less convergence time in the authentication process. The convergence is identified using the classification of p based on the occurrence of the $\rho_{p}$ and $\rho_{s}$. Following the classification process, N(p) for ${\rho (p|s}_{i})$, i∈c or ${\rho (p}_{i}|s)$, the converging time is identified in forehand, restricting in breaches in sharing and shared data tampering. Therefore, the identification based on p and N(p) helps to divide the authentication for $\rho_{a}>\frac{R}{Q}$ and $\rho_{a}<\frac{R}{Q}$ instances. The verification and authentication observed for the above conditions are different, without generating different point and probabilities. Here, detection of p segregates the authentication process for sequential and concurrent instances as 1 to p and p to c without requiring a new hash or verification procedure. As the number of convergence increases, the concurrency is increased without requiring additional computation steps. Therefore, the probabilistic classification of $\overset{c}{\underset{i=1}{\prod }}\rho (p_{i}|s)$ and $\overset{c}{\underset{i=1}{\prod }}\rho (p|s_{i})$ for N(p) achieves less convergence in the proposed security system (refer to Figure 7). In Table 5, the comparative analysis results are tabulated. 

From Table 5, it is seen that the proposed security system is capable of achieving better performance by reducing the response time and increasing the ratio of successful sharing through trust-based validations. In the authentication process, the computation and converging time are found to be less since the instances of sharing are segregated based on p.

As in Table 5 and in Figure 8, the proposed security system achieves a very high performance for analyzing various attacks. The better performance is achieved by consuming low response time, less computation time and reduced converging time. As opposite, it achieves a high successful sharing rate. 

## 6. Conclusions

This paper introduced a blockchain and distributed ledger-based improved biomedical security system for improving the privacy and security of EHRs. This security system relies on the blockchain paradigm for providing trust validation through linear decision-making. The authentication of EHRs is preceded using classification-based learning for identifying sequential and concurrent sharing. The process is focused on both user-level and sharing-level security and privacy of the biomedical systems. The classification of sharing instances helps to reduce the complex and overloaded computations in the authentication process with less computation time. The blockchain technology coupled with this process helps to share trust-related information and differentiate the sharing based on classification instances. The experimental analysis of the proposed security system shows that it is capable of increasing the sharing ratio by 8.077% and 7.03% for sharing time and records, respectively. It also achieves 20.11% less response time compared to the other methods. In the case of authentication, the proposed system confines computation and convergence time by 10.26% and 12.31%.

## Figures and Tables

**Figure 1 healthcare-08-00243-f001:**
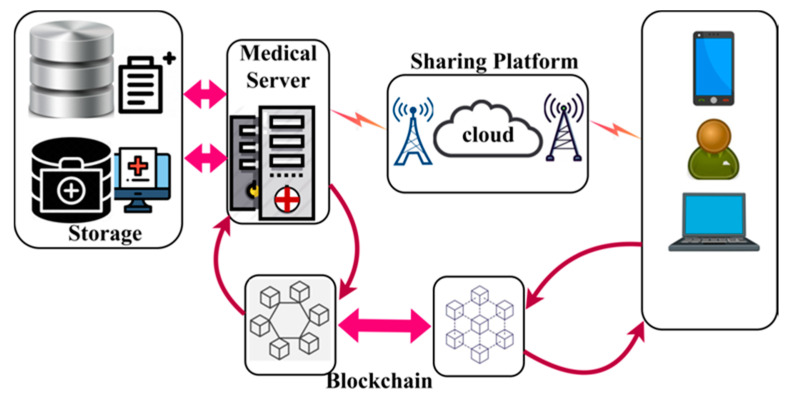
Biomedical Security System with blockchain.

**Figure 2 healthcare-08-00243-f002:**
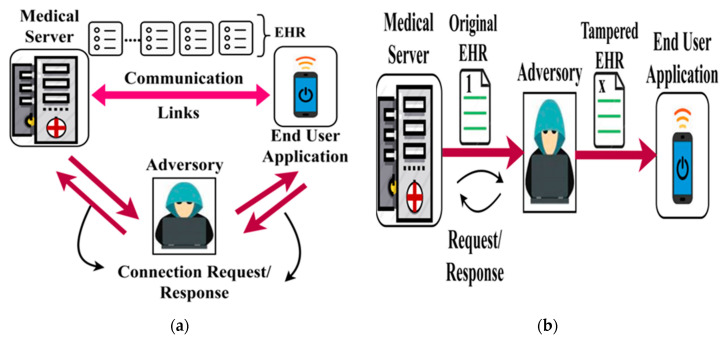
(**a**) Man-in-middle attack. (**b**) Data tampering attack. (**c**) Server-client based blockchain technology.

**Figure 3 healthcare-08-00243-f003:**
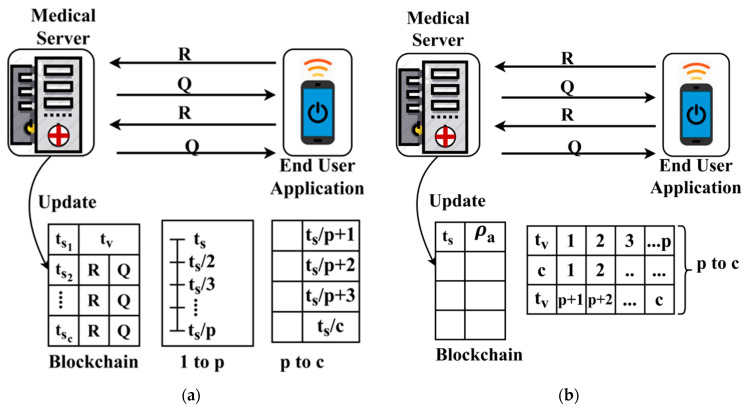
(**a**) Sequential update (1 to p), (**b**) concurrent update (p to c).

**Figure 4 healthcare-08-00243-f004:**
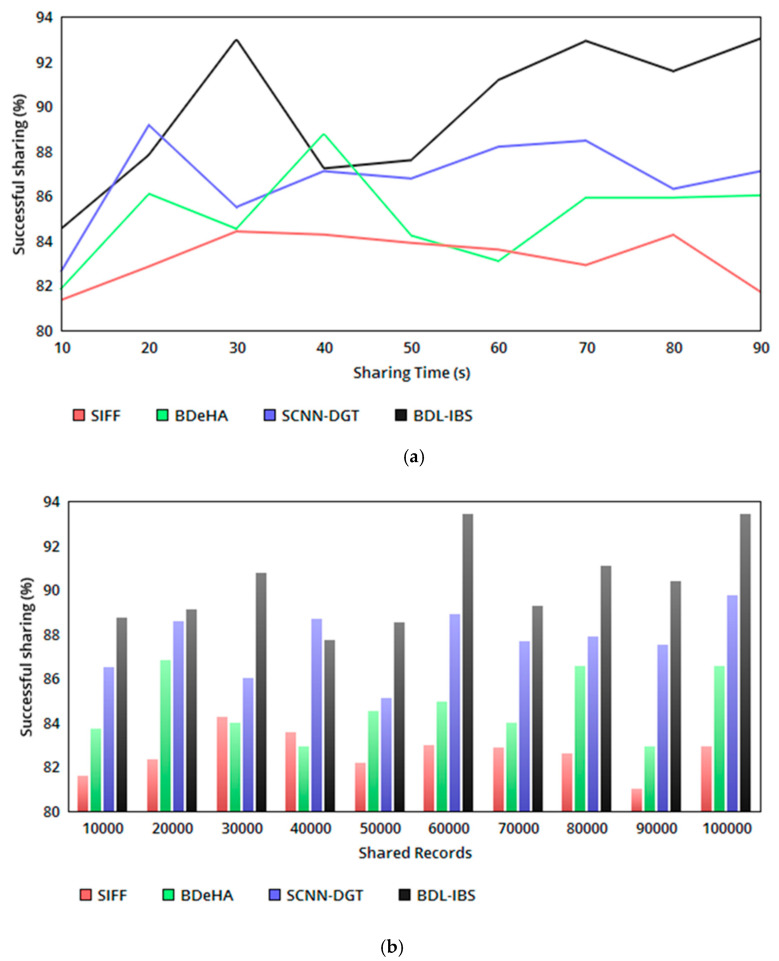
(**a**) Successful sharing ratio versus sharing time. (**b**) Successful sharing ratio versus shared records.

**Figure 5 healthcare-08-00243-f005:**
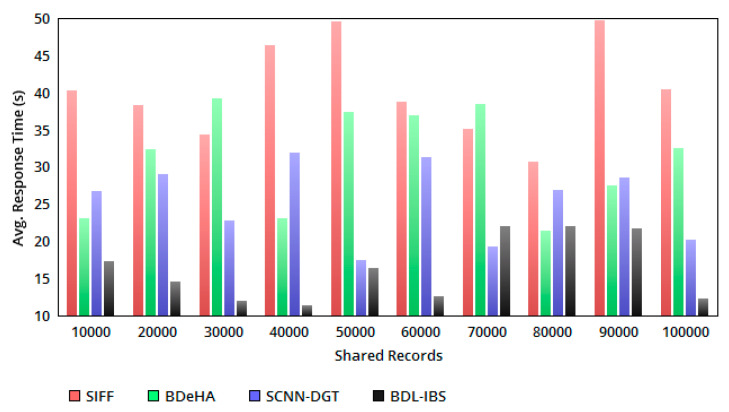
Avg. response time versus shared records.

**Figure 6 healthcare-08-00243-f006:**
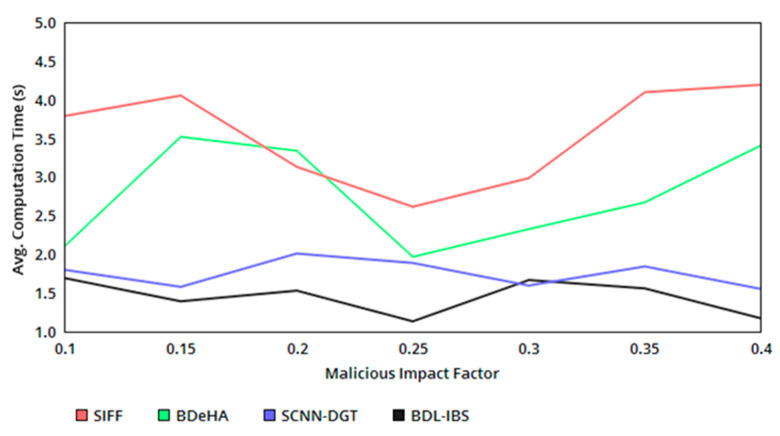
Avg. computation time versus malicious impact factor.

**Figure 7 healthcare-08-00243-f007:**
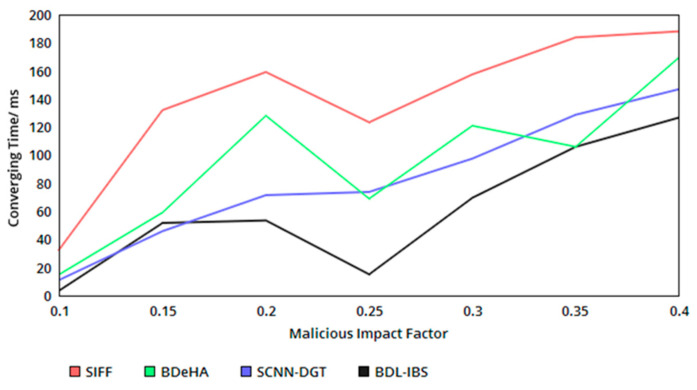
Converging time versus malicious impact factor.

**Figure 8 healthcare-08-00243-f008:**
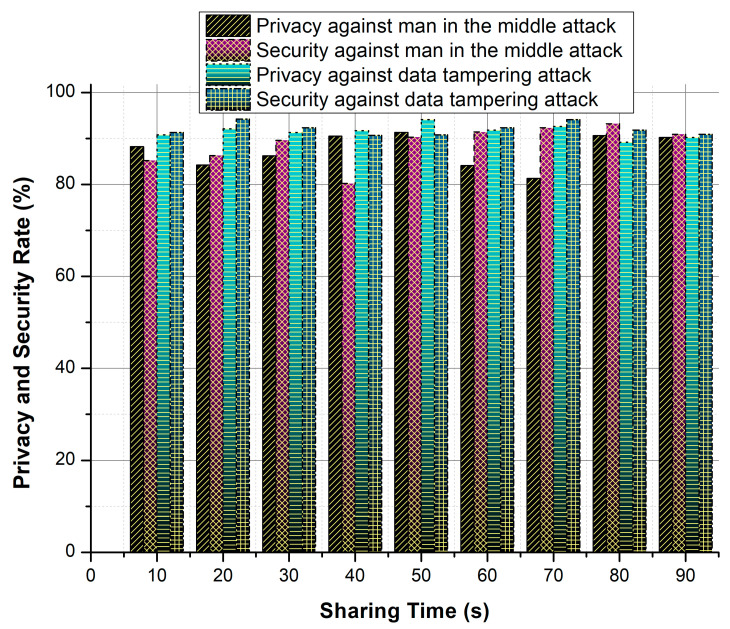
Privacy and security ratio against attacks.

**Table 1 healthcare-08-00243-t001:** Blockchain Entries.

Field	Description
Name/Id	User Name/Login Credential
*Q*	Query Request
*R*	Response
*c*	Count of EHR shared
$t_{s}$	Sharing Time
$t_{v}$	Validity Time
τ	Trust Factor

**Table 2 healthcare-08-00243-t002:** Decision conditions.

Condition	Description	Solution
$\overset{c}{\underset{i=1}{\sum }}{\left(\frac{R}{Q}-\rho_{a}\right)}_{i}<\overset{c}{\underset{i=1}{\sum }}{\left(\frac{R}{Q}-\rho_{a}\right)}_{i-1}$	Current trust is less than the previous trust in any of the instance of the sequence	Pause sharing until the next update is received
p<c	The actual share count is high that the identified point	Continue sharing until p=c is reached.
p≥c	The identified point is greater than the shared EHRs.	Halt EHR sharing
$\bar{\tau }=\tau \left(t_{v}\right)$	Sequence trust is the same as the instance trust value computed	Not feasible until c=1
$\bar{\tau }<\tau \left(t_{v}\right)$	Sequence trust is high that the instance trust value	Halt EHR sharing

**Table 3 healthcare-08-00243-t003:** Records Classified under Table 1 Condition.

Sharing Time (s)	Condition 1	Condition 2	Conditions 3/5	Condition 4
10	374	7152	36	0
20	718	8089	44	0
30	433	8452	17	0
40	847	7843	82	0
50	622	8741	139	0
60	249	9527	86	0
70	506	8719	152	7
80	521	9013	127	0
90	362	9486	92	0

**Table 4 healthcare-08-00243-t004:** $\rho_{s}$ and $\rho_{p}$ and Complexity.

p	$\rho_{s}$	$\rho_{p}$	$t_{s}(s)$	Complexity	*c*
1	0.59	0.38	14.72	0.12	380
2	0.74	0.23	37.49	0.069	887
3	0.64	0.33	46.44	0.052	1028
4	0.43	0.52	78.37	0.083	1849
5	0.74	0.24	78.19	0.064	2053
6	0.69	0.29	88.43	0.087	3188
7	0.82	0.15	79.77	0.042	2207
8	0.54	0.43	69.29	0.103	1352
9	0.59	0.38	76.13	0.096	1511
10	0.73	0.26	84.22	0.067	2733

**Table 5 healthcare-08-00243-t005:** Comparative Analysis.

Metrics	SIFF	BDeHA	SCNN-DGT	BDL-IBS
Successful sharing (%)	82.92	86.55	89.76	93.44
Avg. Response Time (s)	40.46	32.56	20.12	12.21
Avg. Computation Time (s)	4.192	3.407	1.552	1.172
Converging Time/ms	188.09	169.43	146.89	126.7

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
