# Peer review of "Enhancing Privacy and Data Security across Healthcare Applications Using Blockchain and Distributed Ledger Concepts"

_healthcare, 2020, doi:10.3390/healthcare8030243_

Round 1
Reviewer 1 Report
The authors introduce blockchain and distributed ledger-based improved biomedical security system for improving the privacy and security of EHRs.
However, I also have some comments on this paper. For security analysis, the authors must list several attack cases in real world and analyze the security of our model based on these assumptions. I keep in my mind that the formal security proof should be given to prove the security for the proposed protocol.
There are some technical problems to improve the model on the security analysis. I recommend to conditional acceptance and the author should revise the paper to follow the formal security proof.
Furthermore, the authors should expand the Section 2. List some suggestions:
- Blockchain: Securing a new health interoperability experienceC Brodersen, B Kalis, C Leong, E Mitchell… - Accenture …, 2016 - truevaluemetrics.org
- Integrating blockchain for data sharing and collaboration in mobile healthcare applications X Liang, J Zhao, S Shetty, J Liu… - 2017 IEEE 28th Annual …, 2017
- Allowing non-identifying information disclosure in citizen opinion evaluation F Buccafurri, L Fotia, G Lax - … on Electronic Government and the Information …, 2013
- Privacy-preserving resource evaluation in social networks F Buccafurri, L Fotia, G Lax - 2012 Tenth Annual International …, 2012
Author Response
The authors introduce blockchain and distributed ledger-based improved biomedical security system for improving the privacy and security of EHRs.
However, I also have some comments on this paper. For security analysis, the authors must list several attack cases in real world and analyze the security of our model based on these assumptions. I keep in my mind that the formal security proof should be given to prove the security for the proposed protocol.
Response: In a man-in-middle attack, the adversary overlaps the end-user to gain access to the HER. This results in sharing health information to an adversary and thus degrading the design of a secure biomedical system which helps to improve the security of the model which we are using.
There are some technical problems to improve the model on the security analysis. I recommend to conditional acceptance and the author should revise the paper to follow the formal security proof.
Response: The corresponding proof has been clearly noted in the Eq(8(a), 8(b)).
Furthermore, the authors should expand the Section 2. List some suggestions:
1. Blockchain: Securing a new health interoperability experienceC Brodersen, B Kalis, C Leong, E Mitchell… - Accenture …, 2016 - truevaluemetrics.org
Integrating blockchain for data sharing and collaboration in mobile healthcare applications X Liang, J Zhao, S Shetty, J Liu… - 2017 IEEE 28th Annual …, 2017
Allowing non-identifying information disclosure in citizen opinion evaluation F Buccafurri, L Fotia, G Lax - … on Electronic Government and the Information …, 2013
Privacy-preserving resource evaluation in social networks F Buccafurri, L Fotia, G Lax - 2012 Tenth Annual International …, 2012
Response: The reference as suggested by the reviewer has been incorporated [36]
Reviewer 2 Report
This paper proposed a new healthcare data sharing system based on blockchain to enhance security and privacy. The proposed BDL-IBS manages the user request's trust factor through blockchain. However, I have a few questions in this system.
First, as a distributed database, the blockchain technology aims to involve multiple users. If the only medical server in the proposed system is the one that uses the blockchain, what is the difference between using the blockchain to manage the trust factor and using the traditional database system?
Second, is the data stored in the blockchain encrypted? Is it possible for an attacker or other user to deliberately lower R/Q values of a specific user by using the validity time and trust factor stored in the blockchain?
Third, how did this system consider the cost of mining and the difficulty of mining in using the blockchain?
Also, some minor modifications are needed to improve readability.
1. A summary description of each chapter should be added at the end of the Introduction.
2. Words such as "Blockchain" and "Block chain" are needed to be unified.
3. The ratio of figure 2-b needs to be extended for better readability.
4. Correction of typos is required (ex: "factor c" -f "factor C" of line 166, etc.
Author Response
This paper proposed a new healthcare data sharing system based on blockchain to enhance security and privacy. The proposed BDL-IBS manages the user request's trust factor through blockchain. However, I have a few questions in this system.
First, as a distributed database, the blockchain technology aims to involve multiple users. If the only medical server in the proposed system is the one that uses the blockchain, what is the difference between using the blockchain to manage the trust factor and using the traditional database system?
Response: This security system relies on the block chain paradigm for providing trust validation through linear decision-making for the purposed of trust factor and using the traditional database system it has been quantified. The authentication of EHR's is preceded using classification-based learning for identifying sequential and concurrent sharing.
Second, is the data stored in the blockchain encrypted? Is it possible for an attacker or other user to deliberately lower R/Q values of a specific user by using the validity time and trust factor stored in the blockchain?
Response: Trust model based on Linear Decision Making: In the trust model, the factors successful access, end-user application to fetch HER. Through conventional communication standards, the end-user application generates a query for accessing HER. The initial authorization for the end-user is provided using login id/ name and password information. This information is validated by the medical server to ensure the reputation of the user. The medical server is associated with the block chain with the following entries and validity time shown in the Table 1.
Third, how did this system consider the cost of mining and the difficulty of mining in using the block chain?
Response: In this process, the block chain updates the trust for the linear and relation, that remains unchanged. Therefore, sharing for varying time and EHR’s follows conditional satisfaction as in Table2, achieving a high successful sharing ratio [Refer to figure 4(a) and 4(b)] helps to optimize cost of mining.
Also, some minor modifications are needed to improve readability.
1. A summary description of each chapter should be added at the end of the Introduction.
Response: It has been updated (However, Block chain refers to the distributed ledger technology which constitutes an innovation in the information recording and sharing without a trusted third party. In this paper, Block chain and Distributed Ledger based Improved Biomedical Security system (BDL-IBS) has been proposed to enhance the privacy and data security across healthcare applications.
Words such as "Blockchain" and "Block chain" are needed to be unified.
Response: It has been updated
3. The ratio of figure 2-b needs to be extended for better readability.
Response: It has been updated
4. Correction of typos is required (ex: "factor c" -f "factor C" of line 166, etc.
Response: It has been updated as per the comments rendered.
Reviewer 3 Report
Summary: This paper presents a blockchain based mechanism for secure sharing of healthcare records and information. The paper presents a blockchain based architecture and analytically evaluates the performance of the feasibility of the architecture.
Detailed Comments:
While the goals and focus of the paper are interesting, the key issues are in the implementation of the research:
- There is a wide array of publications and prior research that discuss the potency and architectures for blockchain based healthcare record sharing. The novelty of the proposed approach does not come through. The rationale for the proposed architecture must be stronger and presented in detail.
- Thought the focus of the architecture is to ensure secure record sharing, the aspect of security discussions and methods to improve security is limited to none in the paper. How does the proposed architecture address security? What are the attack models? How does it address each of those attacks? And relating to (1), why blockchain for such attack models?
- The discussion of the mechanism of the approaches are very mathematical. While it is appreciated that there is analytical support to the design, the key research questions need to be answered clearly in descriptions and the mathematical proof can follow as evidences of evaluation.
- The evaluation section is fuzzy. The rationale for the metrics chosen are unclear. What is the relation between the metrics, the measures and the goals of the project?
Writing:
The paper requires a significant revision for grammar, typos and unnecessary captitalization avoiding.
Author Response
Summary: This paper presents a blockchain based mechanism for secure sharing of healthcare records and information. The paper presents a blockchain based architecture and analytically evaluates the performance of the feasibility of the architecture.
Detailed Comments:
While the goals and focus of the paper are interesting, the key issues are in the implementation of the research:
There is a wide array of publications and prior research that discuss the potency and architectures for blockchain based healthcare record sharing. The novelty of the proposed approach does not come through. The rationale for the proposed architecture must be stronger and presented in detail.
Response: Block chain refers to the distributed ledger technology which constitutes an innovation in the information recording and sharing without a trusted third party. In this paper, Block chain and Distributed Ledger based Improved Biomedical Security system (BDL-IBS) has been proposed to enhance the privacy and data security across healthcare applications
Thought the focus of the architecture is to ensure secure record sharing, the aspect of security discussions and methods to improve security is limited to none in the paper. How does the proposed architecture address security? What are the attack models? How does it address each of those attacks? And relating to (1), why blockchain for such attack models?
Response: It is either modifies the actual data/ tracks the communication through the HER information. Figure 2(a) and 2(b) portray the representation of the man-in-middle and data tampering attacks over the EHR.
The discussion of the mechanism of the approaches are very mathematical. While it is appreciated that there is analytical support to the design, the key research questions need to be answered clearly in descriptions and the mathematical proof can follow as evidences of evaluation.
Response: The mathematical proof has been derived between Eq(8a) to 9(c).
The evaluation section is fuzzy. The rationale for the metrics chosen are unclear. What is the relation between the metrics, the measures and the goals of the project?
Response: The initial authorization for the end-user is provided using login id/ name and password information. This information is validated by the medical server to ensure the reputation of the user based on the metrics as shown in the Table.1.
Writing:
The paper requires a significant revision for grammar, typos and unnecessary captitalization avoiding.
Response: Proof reading has been done.
Round 2
Reviewer 2 Report
For all comments, the author has revised the content of the paper appropriately.
Author Response
Thank you very much for sharing positive feedback towards our revised manuscript
Reviewer 3 Report
I thank the authors for clarifying some aspects of the work. However, as pointed out before, the attack model description, its impact on the design, the solution and evaluation of the same, are not convincing, yet. The consideration of man-in-the-middle attack in this work seems very simplistic -- raises the question as to why regular 2-layer authentication schemes will not work for the same.
I suggest the authors to consider an array of attack models and discuss how the proposed architecture works to address the same. The evaluation can either be simulatory or trace based, if real experiments are not accessible at this stage.
Another suggestion is to tie the mathematical explanations strongly with the research descriptions. Most of the mathematical analysis parts are very challenging to read, even for a person in this area of research. The paper needs better organization.
Author Response
I thank the authors for clarifying some aspects of the work. However, as pointed out before, the attack model description, its impact on the design, the solution and evaluation of the same, are not convincing, yet. The consideration of man-in-the-middle attack in this work seems very simplistic -- raises the question as to why regular 2-layer authentication schemes will not work for the same.
Ans: However, Block chain refers to the distributed ledger technology which constitutes an innovation in the information recording and sharing without a trusted third party. In this paper, Block chain and Distributed Ledger based Improved Biomedical Security system (BDL-IBS) has been proposed to enhance the privacy and data security across healthcare applications. In a man-in-middle attack, the adversary overlaps the end-user to gain access to the HER. This results in sharing health information to an adversary and thus degrading the design of a secure biomedical system. In the case of a data tempering attack, the adversary breaches HER from any node communicating with the biomedical system. It is either modifies the actual data/ tracks the communication through the HER information. Figure 2(a) and 2(b) portray the representation of the man-in-middle and data tampering attacks over the EHR. Trust model based on Linear Decision Making: In the trust model, the factors successful access, end-user application to fetch HER. Through conventional communication standards, the end-user application generates a query for accessing HER. The initial authorization for the end-user is provided using login id/ name and password information. This information is validated by the medical server to ensure the reputation of the user. The medical server is associated with the block chain with the following entries, as in Table 1.
I suggest the authors to consider an array of attack models and discuss how the proposed architecture works to address the same. The evaluation can either be simulatory or trace based, if real experiments are not accessible at this stage.
Ans:In this process, the block chain updates the trust for the linear and relation, that remains unchanged. Therefore, sharing for varying time and EHR’s follows conditional satisfaction as in Table2, achieving a high successful sharing ratio [Refer to figure 4(a) and 4(b)].
Another suggestion is to tie the mathematical explanations strongly with the research descriptions. Most of the mathematical analysis parts are very challenging to read, even for a person in this area of research. The paper needs better organization.
Ans: Ans: The above likelihood normalization of helps to classify condition or condition. This helps to decide between sequential and concurrent authentication procedure through the same concentric point from the fluctuating sequence of . The normalization identifies precise in the series of such that follows sequential authentication whereas the previous occurrence relies on random concurrent security measures. Here, the priority of authentication is initiated from the first occurrence of of as determined by . For all the first occurrence of and , the sequence follows or and [as in equation (6)]. Using this sequence and concurrency, the authentication is presented as follows. In two cases, the occurrence of the sequence and concurrency observed is discussed below.
In addition I have attached the equation in the word file. Please
